# Levels and Patterns of Physical Activity and Sedentary Behaviour of Primary School Learners in Lagos State, Nigeria

**DOI:** 10.3390/ijerph191710745

**Published:** 2022-08-29

**Authors:** Olusegun Olatunji Ojedoyin, Oladapo Michael Olagbegi, Thayananthee Nadasan, Pragashnie Govender

**Affiliations:** 1Discipline of Physiotherapy, School of Health Sciences, University of KwaZulu-Natal, Durban 4000, South Africa; 2Harvard Medical Rehabilitation Hospital, Ikorodu, Ikorodu P.O. Box 3143, Lagos, Nigeria; 3Discipline of Occupational Therapy, School of Health Sciences, University of KwaZulu-Natal, Durban 4000, South Africa

**Keywords:** child, exercise, non-communicable diseases, schools, social class, Nigeria

## Abstract

Background: Physical activity (PA) and sedentary behaviour (SB) assessment in children is critical for the prevention of noncommunicable diseases. African studies examining PA and SB of primary school-age children are few. This study investigated PA, SB levels and their correlates among primary school children in Lagos, Nigeria. Method: In a cross-sectional study of 733 learners, their self-reported PA and SB were assessed using the Children PA Questionnaire (CPAQ) (6–9 years age category) and Youth Activity Profile (YAP) (10–12 years age category) while pedometers were used for objective PA and SB assessment, and socioeconomic status (SES) index were measured using a structured questionnaire. Standardised procedures were used for anthropometric and cardiovascular measures. Results Based on CPAQ, 87.5% and 100% of the learners aged 6–9 years met the recommended PA and SB guidelines, respectively which were lower with pedometers (72.8% and 87.3%). The proportion of boys aged 6–9 years who met the guidelines for PA and SB (using pedometer) was significantly higher than that of the girls(PA: 80.7% vs. 64%, *p* = 0.018; SB: 94% vs. 80%, *p* = 0.008). Self-reported PA was positively associated with age (CPAQ: B = 455.39, *p* < 0.001; YAP: B = 1.638, *p* = 0.009) and negatively with SES (CPAQ: B = −201.39, *p* < 0.001; YAP: B = −1.000, *p* < 0.001). Objective PA was positively associated with waist to hip ratio(WHR) (6–9 years: B = 66090.24, *p* = 0.032) and negatively with sex (6–9 years: B = −5533.41, *p* = 0.027) and hip circumference (10–12 years: B = −1269.13, *p* = 0.017). SB was associated with SES in learners aged 10–12 years (B = −0.282, *p* = 0.003).Conclusion: High SES is a major predictor of reduced PA among these cohort of learners.

## 1. Introduction

Reducing sedentary behaviour (SB) is clearly a good step towards promoting physical activity (PA) among children. Physical inactivity, on the other hand has been described as “the biggest public health problem of the 21st century [1,2]. Meeting PA guidelines among school aged children and adolescents is known to be positively associated with physical, psychological, social, and cognitive health indicators [3], while physical inactivity, defined as not meeting PA guidelines, is associated with adverse physical, mental, social outcomes [4,5]. PA improvements reduce overweight and obesity. Importantly, high levels of SB and insufficient levels of PA increase childhood and adolescent obesity, as well as the risk of morbidity and mortality, from cardiovascular and metabolic diseases (e.g., type 2 diabetes) among adolescents [6]. Children who engage in PA are more likely to maintain a healthy weight, have stronger, healthier bones and muscles, and have better heart and brain health. In addition, they are likely to perform better academically, especially in the subjects of mathematics, reading, and writing [7,8,9]. Children who participate in a large amount of PA have a lower long-term risk of developing diabetes, heart disease, a few types of cancer, and anxiety [10]. 

PA has been defined as any bodily movement produced by skeletal muscle that requires energy expenditure [6,11]. PA can be achieved by walking, cycling, and participating in any type of sporting activity. PA can also be in form of house chores, such as dish washing, sweeping and other similar tasks [12]. All forms of PA provide health benefits, and international PA guidelines for primary school aged children are available. For instance, the World Health Organisation (WHO) recommends that “*PA for children aged 5–17 years, includes play, games, sports, transportation, recreation, physical education, or planned exercise, in the context of family, school, and community activities. To improve cardiorespiratory and muscular fitness, bone health, cardiovascular and metabolic health biomarkers and reduced symptoms of anxiety and depression, the WHO further recommends that the following children and adolescents aged 5–17 years old should engage in at least 60 min of moderate-to-vigorous-intensity aerobic activity daily, as well as vigorous-intensity activities, including those that strengthen muscle and bone (at least three times per week)”* [13].

SB is a different paradigm from PA which is defined as any waking behaviour characterized by an energy expenditure ≤1.5 metabolic equivalents, such as sitting, reclining, or lying down [14]. It has been shown that high levels of continuous sedentary behaviour (such as sitting for long periods of time) are associated with abnormal glucose metabolism and cardiometabolic morbidity [15,16]. Consequently, we can infer that reducing SB through the promotion of incremental PA such as from standing to stair climbing, can help individuals to increase their levels of PA and achieve optimal health. For instance, the WHO recommends that for children aged 5–17 years should limit the amount of time spent being sedentary, particularly recreational screen time [17]. Some known national guidelines recommend limiting sedentary recreational screen time to not more than 2 h daily and recommend intermittent breaking up of long periods of sitting [18,19]. Similarly, there are widespread worries about the increased incidence of lifestyle related non-communicable diseases (NCDs). Type 2 diabetes and other NCDs are becoming more common around the globe and affect people of all ages, including children. A declining quality of life and rising healthcare costs are the ultimate consequences. Improved PA has been shown to successfully prevent a number of NCDs making it a crucial focus for health promotion [11,20].

Despite the numerous health benefits of PA promoting daily activities, such as walking and cycling, their popularity is declining in many countries because of urbanisation. Our roads are generally not safe enough because of lack of policies that prioritize pedestrians and cyclists’ access to guarantee their safety, particularly for school children [21]. With increasing urbanization, and more people moving to the cities, governments have a special responsibility to improve urban design with the aim of encouraging school children to use active transportation for promotion of public health [22,23]. In addition, school-based policy on PA is important to create a more active lifestyle for school children. PA is important for people of all ages and should be incorporated into a variety of daily programmes. Adults workplace seems a major setting to be physically active and reduce SB, whereas, the school children may lack opportunity for PA if school and environments do not promote such [24]. 

Currently, Nigeria, a member of WHO, lacks a National PA policy for all age groups as opposed to the WHO’s recommendations [25,26,27], although Physical and Health Education (PHE) programmes have been incorporated into school curriculum for primary school children. However, it remains to be seen how far this can help improve the children’s PA. Furthermore, there is dearth of baseline PA data for Nigerian primary school aged children. A previous study by Ajayi et al. among primary school students in Lagos, Nigeria, found that school-based PHE programmes did not meet recommended national and state guidelines. The authors further observed significant prevalence of overweight/obesity among the school children which was attributed to a lack of optimal PA [28]. Hence there’s need for sustained efforts to improve learners PA in order to prevent future health risks. With this background in mind, the purpose of this study was to examine the levels of PA and SB among primary school children in Lagos State, Nigeria. The secondary objective of the study was to examine the sociodemographic, anthropometric, and cardiovascular correlates of PA among this cohort of primary school children. Specifically the study was aimed at addressing the following research questions:What would be the baseline PA and SB data of primary school children in Lagos state, Nigeria.Are the PA and SB scores and levels of primary school children in Lagos State, Nigeria, statistically comparable across gender?What would be the levels of PA among primary school children in Lagos State, Nigeria.Is PA associated with some selected socioeconomic, cardiovascular, and anthropometric variables of primary school children in Lagos State, Nigeria.

## 2. Methods

### 2.1. Study Design

This study utilized a quantitative cross-sectional design. 

### 2.2. Participants

The study participants were 733 primary school learners (393 boys, 340 girls), aged 6–12 years. They were randomly selected from 40 public and private primary schools of Lagos State, Nigeria. To ensure that a diverse mix of socio-economic backgrounds were chosen, computer-generated random numbers were used to select schools, after codes were assigned to public and private schools across all local governments in the state. 

### 2.3. Sampling Technique and Sample Size Calculation

Primary school children were recruited for the study using non-probability purposive sampling adopted to recruit study participants. This was contingent on the availability of parental or guardian consent and child’s assent. The formula for cross-sectional studies described by Charan and Biswas [29] was used to calculate and determine a minimum sample size of 384 participants for the study. 

#### 2.3.1. Inclusion and Exclusion Criteria 

A child was included in the study if he/she was enrolled in the selected primary schools in 2019/2020 academic session and were between ages 6 and12 years. Children with any medical history (reported by parents or teachers) that would not allow them to cope with the physical demands of the study or children with any form of disability and those who are unable to walk, or those Children who have reached the stage of puberty were excluded. The Modified Maturity Offset Prediction Equations validated by Koziel et al. [30] were used to screen out those with early signs of puberty.
Maturity offset (years) (Girls) = −7.709133 + (0.0042232 × (age × stature)
Maturity offset (years) (Boys) = −7.999994 + (0.0036124 × (age × stature)
Maturity age = (Chronologic Age + Maturity Offset) years

#### 2.3.2. Ethical Considerations

Ethical approvals for the study were granted by the **Biomedical Research Ethical Committee (BREC)** of the University of KwaZulu-Natal (Reference number **BREC/00000523/2019)**, and the **Health Research Ethical Committee (HREC)** of Lagos State University Teaching Hospital (Reference number **LREC/06/10/1331**). Gatekeepers’ permissions were also obtained from the Head of Service and Permanent Secretary, the Ministry of Health, and the Chairman of State Universal Basic Education Board (SUBEB), Lagos State, Nigeria. 

#### 2.3.3. Data Collection Process

Information letters and informed consent forms were sent to the Headmaster/Headmistresses of the selected schools following ethical approval and gatekeeper permission. These were explained in English and Yoruba languages to provide information sessions with the children concerning the study and to inform them that their participation is voluntary. Potential participants were children who returned duly signed informed consent letters from their parents. Only eligible learners who agreed to sign a minor assent form were eventually recruited for the study. Biographical questionnaire was used to collect data on age, gender, and class of study.

#### 2.3.4. Socio-Economic Status (SES) 

Participants’ SES was assessed by asking them to fill a questionnaire. The questions were adopted from a similar instrument used in a South African study [31]. The questionnaire items covered household-level living standards, such as infrastructure and housing characteristics (house type, number of bedrooms, type of toilet and access to indoor water, indoor toilet/bathroom, and electricity) as well as questions about ownership of three durable assets (presence of a working refrigerator, washing machine, and car). The dichotomised items (0 = poor quality, not available; 1 = higher quality, available) were summed to create an overall SES index, with higher scores reflecting higher SES. 

#### 2.3.5. Cardiovascular Measures (Blood Pressure and Heart Rate)

In the sitting position, children’s resting systolic and diastolic blood pressure (SBP and DBP) and heart rate (HR) were measured using an electronic Blood Pressure (BP) monitor (Sam vine, USA) [32]. Following five minutes of sitting at rest, measurements were taken five times at two-minute intervals, and the average value of the last three measurements was used in the analysis. 

#### 2.3.6. Anthropometric Measurements

During body weight measurements, the child stood on the scale for five seconds with feet hip-width apart. Bodyweight was measured to the nearest 0.1 kg. The weighing scale was calibrated using repeatability and eccentricity tests before each week’s body weight assessment [33].

To measure standing height (stature), the participant stood with heels together and heels touching the base of the stadiometer. The body mass index (BMI) was calculated using height and weight (BMI = body weight/height^2^). Waist circumference was measured to the nearest 0.1 cm with an inelastic tape measure and the umbilicus as a reference point [34].

The hip circumference (HC) was measured using the same tape with the most protruding point on the child’s buttocks serving as the reference point. The WHR was calculated from these values.

#### 2.3.7. Self-Reported PA Assessment

**The YAP** (Youth activity profile) was administered to participants aged 10–12 years. A total of **190 (male = 101, female = 89)** learners were assessed using YAP. It contains 15 total items, each scored on a 5-point scale. The first five items assessed PA at school by enquiring about behaviour in different school time, such as transportation to and from school, physical education (PE), recess, and lunch. The next five questions inquired about PA levels at home. Three questions asked how many days they were active before school, after school, and weeknights. Answers were chosen from 0 days at the minimal and 4 to 5 days at the most. Two additional questions enquired more specifically about the length of time they spend on physical activity on Saturday and Sunday. The last five items included time spent in different sedentary behaviours’ (TV time, video games, 30 min computer time, Phone/Text time and overall siting time). Separate behavioural indices were computed from the three subscales of the YAP. The three indices are PA at School Index (PASI), PA at Home Index (PAHI), and SB Index (SBI). The score for each index was calculated by averaging the responses from the five equally weighted items in each category. A higher overall activity score is reflective of a higher level of physical activity (or a greater amount of sedentary behaviour in the case of the SBI). YAP scores could not be categorised for levels of PA because information for its categorisation was not readily available. Saint-Maurice et al. found a low to moderate correlation between YAP PA scores and estimates from an accelerometer (r = 0.19–0.58). They also reported a high correlation between an estimates of time spent in SB for both instruments [35].

The Children Physical Activity Questionnaire (**CPAQ**) was used to assess the PA of 543 (male = 292, female = 251) children aged 6–9 years. It contains 48 total items for week days and weekends activities. Nor Ani et al. reported a test-retest reliability score of 0.55 to 0.68 for CPAQ (*p* < 0.001) and concluded that the CPAQ exhibits good capacity to appropriately identify children and has reasonable validity in assessing moderate physical activity [36]. The interviewer administered the CPAQ which assessed three domains as follows: activities in relation to school; organized activities; and unorganized activities. Children recalled activities from the past week and reported habitual PA, including school activities, all forms of transportation, physical education sessions, activities in sports and other clubs, leisure time activities and work activities in the past 7 days. Therefore, the CPAQ assessed mode, frequency, and duration of PA and sedentary activities throughout all domains over the past 7 days.

For CPAQ, the weekly and weekend scores were summed to calculate total PA scores. Total PA scores were also unbundled into 7 namely, **sports PA weekday, sports PA weekend, leisure PA weekday, leisure PA weekend, activity at school weekday, total weekday PA and finally total weekend PA.**

The WHO recommend international guidelines of engaging in at least 60 min of moderate-to-vigorous physical activity (MVPA) daily, based on self-reported PA [37].

#### 2.3.8. Objective PA Assessment with Pedometers

Pedometers (*Omron Healthcare, South Africa*) were strapped to each of the participants for 7 consecutive days. A pedometer is a small device that counts the number of steps you take. Pedometers are portable electronic or electromechanical devices that count a person’s steps and estimate the distance a person walks over a period. A total of **201 (male = 109, female = 92)** primary school learners were administered pedometers. Instructions were given to them not to remove it until when they were going to the bathroom and return the device to the body almost immediately. This pedometer takes readings for consecutive seven days and saved on the pedometer for another 14 days. The readings for the last consecutive seven days were taken and recorded in the data sheet as daily step counts for the individual. The addition of all the readings for the consecutive seven days were done and referred to as **total weekly step counts**, while the total weekly step counts for the individual and divided by seven days is referred **to** as **average step counts per day**.

WHO guidelines recommend up to 60 min of activity that is of at least moderate to vigorous intensity [37]. Multiplying 60 min by 100 steps/minute results in 6000 steps, However, SB a ‘sedentary’ level of 5000 steps/day is the benchmark.

In a review by Tudor- Locke and others, the updated international literature indicates that we can expect among children, boys to average 12,000 to 16,000 steps/day and girls to average 10,000 to 13,000 steps/day; and adolescents to steadily decrease steps/day until approximately 8000–9000 steps/day and having less than 5000/day is sedentary [38].

### 2.4. Data Analysis

Statistical analyses were performed using the SPSS software, version 27 (IBM, Armonk, NY, USA). Data cleaning procedures were carried out to ensure its accuracy. Shapiro Wilk normality test revealed that the data were skewed, and not normally distributed, consequently, non-parametric tests were used to compare continuous variables. Frequencies, percentages, median and quartiles were used to summarise data descriptively for descriptive summary of data. Mann Whitney U test was used to compare participants’ median scores of sociodemographic, anthropometric, and cardiovascular variables across gender. The non-parametric Quade Analysis of Covariance (ANCOVA) was computed to compare the median PA scores (PA weekday and PA weekend) across gender while controlling for variables that were significantly different in the Mann Whitney U test. The relationships of PA and SB with anthropometric and cardiovascular parameters and socio-economic index were examined using Spearman rank Correlation Coefficient. To determine the predictors of self-reported PA and objective PA with pedometers, variables that showed significant correlations with total PA were fed into the multivariate linear regression models. Separate models 1, 2, 3, 4 and 5 were computed using total PA (from CPAQ), total PA (from YAP) and total step counts (from pedometer separately for participants aged 6–9 years, and those of 10–12 years) as the dependent variables, respectively. The level of significance was set at *p* < 0.05.

## 3. Results

### 3.1. General Baseline Data (Research Question 1)

A total of **733** primary school children aged 6–12 years participated in the study, **393 (53.6%)** were males, and the remaining **340 (46.4%)** were females. Their median age, height, weight, BMI, hip circumference, waist circumference and waist hip ratio were 9.00 (IQR: 8.00, 10.00) years, 1.30 (1.23, 1.36) m, 27.00 (24.00, 30.00) kg, 16.40 (15.30, 17.70) kg/m2, 64.00 (60.00, 68.00) cm, 57.00 (55.00, 60.00) cm, and 0.89 (0.86, 0.93) respectively. The median SES index score of the learners was 6.00 (4.20, 7.20) while their median SBP, DBP and HR were 117.00 (108.00, 129.00) mmHg, 80 (68.00, 98) mmHg, and 97.00 (82.00, 110.00) Beats/minute, respectively.

Comparison of male and female anthropometric, cardiovascular, and socio-economic variables.

The anthropometric, cardiovascular, and socio-economic index of male and female children are compared and presented in Table 1. Boys and girls aged 6–9 years were statistically comparable in their ages, BMI, height, weight, HC, WC, WHR, SES index, SBP, DBP and HR (*p* < 0.05). Boys in the age group 10–12 years had significantly highest SES index median score (*p* = 0.043), while the girls had significantly higher heart rate median score than the boys (*p* < 0.05) in the same age category. Other variables were not significantly different from each other across gender (*p* > 0.05).

### 3.2. Comparison of Participants PA and SB Scores across Gender (Research Question 2)

To address research question 2, participants’ PA and SB scores were compared across gender using Quade ANCOVA while controlling for SES index and HR (covariates) and the results are summarised in Table 2. There was no significant gender difference in the self-reported PA and SB scores among learners aged 6–9 years (*p* > 0.05), The self-reported PA scores of learners aged 10–12 years were not significantly different between boys and girls (*p* > 0.05), while the boys had significantly higher median average step counts (*p* = 0.017) and the total weekly step counts (*p* = 0.017) than the girls for age group 6–9 years, both measures were not significantly different across gender in the age group 10–12 years.

### 3.3. Levels of PA and SB (Research Question 3)

To address research question 3, PA and SB scores were categorized to obtain levels of PA and SB as recommended in literature. Based on the self-reported CPAQ scores categories, 87.5% and 100% of the learners aged 6–9 years met the recommended PA and SB guidelines, respectively. On the other hand, based on the pedometer’s average step counts, it revealed that, overall, 72.8% and 87.3% of learners aged 6–9 years were found to have met the PA and SB guidelines, respectively. In the older 10–12-year-old group, 67.4% and 93% met the PA and SB guidelines, respectively for pedometer, as YAP scores could not be categorised for levels of self-reported PA for children aged 10–12 years because of lack of data for categorisation. Participants’ levels of physical activity and sedentary behaviour were further compared across gender and results are presented in Table 3.

The proportions of boys and girls who met PA guidelines (based on self-reported CPAQ) for children aged 6 to 9 years were not significantly different (*p* > 0.05). However, when PA levels were determined using objectively measured step counts from pedometers, a significantly higher proportion of boys (80.7% vs. 64.0%, *p* = 0.018) met the PA guidelines compared to girls. The proportions of boys and girls who met PA guidelines using objectively measured step counts from pedometers were not significantly different in older children aged 10–12 years (*p* > 0.05).

The results for SB levels followed the same pattern as the results for PA levels, with a significant gender difference in SB levels determined using objectively measured step counts only found in the group aged 6–9 years. The proportion of boys who met the SB guidelines was significantly higher than that of girls (94.0% vs. 80.0%, *p* = 0.008).

Association between participants PA, SB and each of anthropometric, cardiovascular, and socio-economic variables

To address research question 4, a correlation matrix for the relationship of PA with participants’ sociodemographic, cardiovascular, and anthropometric variables was computed and the results are presented in Table 4. Self-reported PA measured with the CPAQ (for participants aged 6 to 9 years) showed significant positive correlations with age (r = 0.323, *p* = 0.001), height (r = 0.092, *p* = 0.039), and a significant negative correlation with SES index (r = −0.318, *p* = 0.001). SB measured using CPAQ did not correlate significantly with any of the variables. For children aged 10–12 years, self-reported PA measured with YAP showed significant positive correlations with age (r = 0.195, *p* = 0.007), BMI (r = 0.280, *p* = 0.001), WT (r = 0.274, *p* = 0.001), WC (r = 0.201, *p* = 0.005), and HC (r = 0.348, *p* = 0.001), and significant negative correlations with WHR (r = −0.250, *p* = 0.001) and SES (r =−0.176, *p* = 0.015). Self-reported SB as measured by YAP had significant positive correlations with BMI (r = 0.154, *p* = 0.034) and negative correlations with HT (r = −0.226, *p* = 0.002) and SES (r = −0.256, *p* < 0.001).

The objectively measured PA using pedometers showed significant positive correlations with WHR (r = 0.157, *p* = 0.049) among participants aged 6–9 years, and significant negative correlations with WT (r = −0.376, *p* = 0.014), and HC (r = −0.498, *p* = 0.001) among those aged 10–12 years.

To further address research question 4, regression models (1–5) were computed to determine the predictors of PA among the two age categories of the participants were presented in Table 5.

For model 1, the ANOVA for the model was significant (F = 56.602, *p* < 0.001) and the model explained 25% of the variances of total PA (measured using CPAQ). The results showed that age (B = 455.39, *p* < 0.001) and SES (B = −201.39, *p* < 0.001) were significantly associated with Total PA (measured using CPAQ). A unit increase in age increased total PA by 455.39 units while a unit increase in SES decreased total PA by 201.39 units.

For model 2, the ANOVA for the model was significant (F = 7.797, *p* < 0.001) and the model explained 23% of the variances of total PA (measured using YAP). The results showed that for children aged 10–12 years, age (B = 1.638, *p* = 0.009) and SES (B = −1.000, *p* < 0.001) were significantly associated with Total PA. A unit increase in age increased total PA by 1.638 units, while a unit increase in SES decreased total PA by 1.000 unit.

For model 3, the ANOVA for the model was significant (F = 4.855, *p* = 0.009), and the model explained 6% of the variances of Total step count (for aged 6–9 years). The results showed that WHR (B = 66,090.24, *p* = 0.032) and Sex (B = −5533.41, *p* = 0.027) were significantly associated with Total step count. A unit increase in WHR increased total step count by 66,090.24 for this age group. With a unit change in sex total step count would decrease by 5533.41 units. The male has higher total step counts.

For model 4, the ANOVA for the model was significant (F = 5.049, *p* = 0.011), and the model explained 21% of the variances of Total step count (for aged 10–12 years). The results showed that HC (B = −1269.13, *p* = 0.017) and total PA (B = 126,016.43, *p* < 0.001) were significantly associated with Total step count. A unit increase in HC decreased total step count by −1269.13 unit, while a unit increase in total step count increased total PA by 126,016.43 unit.

For model 5, the ANOVA for the model was significant (F = 5.307, *p* = 0.002), and the model explained 7.9% of the variances of sedentary behaviour. The result showed that SES (B = −0.282, *p* = 0.003) was significantly associated with total SB scores (measured using YAP). A unit increase in SES decreased SB scores (measured using YAP) by 0.282 units.

## 4. Discussion

The primary objective of this study was to describe and quantify the levels of PA and SB among primary school children in Lagos State, Nigeria, while the secondary objective of the study was to examine the sociodemographic, anthropometric, and cardiovascular correlates of PA among the study participants in this cohort of primary school children.

Participants among the 10–12 years, the boys in the 10–12 years age cohort had significantly higher SES than girls. The result suggests that children from low socioeconomic background participate in activities that promotes active lifestyles. In a study by Gerber et al. among the marginalised black communities of South Africa, they found that children of the lowest SES accumulated higher levels of device-based moderate to vigorous PA, and that peers from the highest SES engaged in more sedentary behaviours [39].

Girls demonstrated higher HR than the boys for the same age cohort 10–12 years. There are several possible explanations for this phenomenon. One possibility is that girls generally have smaller hearts than boys, and thus their heart rate is naturally higher. Another reason is that those girls may be engaged in activities that require more energy and thus lead to a higher heart rate. Additionally, girls may have higher levels of hormones that can lead to an increased heart rate [40,41].

This study also showed that the boys had significantly higher median average step counts and the total weekly step counts than the girls for age group 6–9 years. The boys in 6–9- and 10–12-years age groups had higher, median average step count than girls. This might be because, boys are known to be generally more active than girls. In a study in South Africa by Minnaar and colleagues among black and Caucasian children, Boys in the age groups 9–11 and 12–14 years are statistically more active than girls of the same age [42]. In another study by Zanevskyy et al. in Europe, they found a higher significant difference between step counts during days of week for boys than girls [43].

However, in this study, for levels of PA and SB, 72.8% of participants met the PA guideline of at least 6000 steps per day, while 80.7% of boys and 64% of girls met the same guideline. In addition, a total of 87.3% of participants met the SB guideline of not more than two hours of screen time per day while 94% of boys and 80% of girls respectively met the same guideline. These values are lower than those of a previous study. In a 2016 study by Minnaar et al. [42] in a rural South African village, boys had higher step counts of 13,081 compared to girls’ 9457 counts. The difference between median values in this study and other studies could be due to environmental factors such as hot weather in Nigeria that discourages walking, inadequate facilities for active transportation and insecurity. Other probable explanations also include the anatomic differences between both sexes. The most apparent sex difference is the wider, broader female pelvis and wider pelvic aperture. This contributes to a greater Q-angle in females and greater hip adduction during knee loading [44]. While these differences likely contribute to sex differences in knee injury incidence, they cannot fully explain the sex differences in pelvic motion [45]. Further studies are needed to identify specific causal factors as we are also aware that there are myriad cultural, behavioural, and physiological factors that may impede girls’ activities. Although the average step count for both the males and females in this study appears to be lower than the international values as reported by Tudor- Locke and colleagues [46]. It is apparent that environmental conditions may have been a factor. In Nigeria, the weather is hot and there is no incentive to walk regularly unlike in temperate weather.

The overall PA level from self-report assessment was 87.5%. Gender stratification showed 85.8% boys, and 89.3% girls met the international guidelines of engaging in at least 60 min of moderate-to-vigorous physical activity (MVPA) per day, while 100% met the SB requirements of maximum of 2 h screen time daily. These results were better than what was reported in Nigeria physical activity report cards for 2013 (58.4% for boys and 29.0% for girls) and 2016 (54.3% for boys and 45.7% for girls) [25,26]. This could be because the present study covered just a section of the Nigerian population and the entire country. Another probable reason is the difference in sample size. A similar study by Oyeyemi and colleagues in 2016, reported 37% of 1006 secondary school adolescents participants in urban north-eastern Nigeria met international guidelines for at least 60 min of moderate-to-vigorous physical activity (MVPA) daily, based on self-reported PA [47]. Overall, there was no gender differences in self-reported PA but boys in the 6–9 years group had significantly higher objective PA (average step count and total weekly step count) than girls. This finding is consistent with Minnaar et al.’s study in a rural South African village, in which prepubertal boys had higher step counts than girls [42]. In the 6–9-year cohort, the percentage of boys who met the recommended guidelines was significantly higher than for the girls, based on the categories of objective measure categories PA. This may be because boys are generally more active and playful than girls at this stage. In a similar study in the Johannesburg area of Gauteng province [48], the authors found that overall, 78% of learners reported meeting the recommended guidelines of at least 60 min per day of MVPA, but girls performed worse than boys, and older children performed worse than younger children [48].

This study also showed self-reported PA for age 6–9 years showed significant positive correlation with SES while self-reported PA for 10–12 years showed significant positive correlations with age, BMI, WT, WC, and HC, and significant negative correlations with WHR, and SES. Objective measured PA showed significant positive correlation with WHR for aged 6–9 years, and significant negative correlations with WT, and HC for aged 10–12 years. This might be because for the two age categories 6–9 years and 10–12 years, they are still at prepuberty years, therefore their activity should increase as their WHR increases, WT and HC decrease. In a previous study by Pojskic and colleagues among primary school children aged 10–14 years in Eastern Europe, higher PA was associated with boys, older age, higher WHR, and lower WC [40]. In a similar study by Mcveigh and colleagues in 2004, South African children with the highest socioeconomic status were more physically active, watched less television, and weighed more. Compared with children from lower educational background, a higher percentage of children with higher socioeconomic status were very active. In the groups with lower socioeconomic status, the percentage of those who exercised little and watched a lot of television was high [49].

This study also found that age and SES were predictors of self-reported PA in the 6–9 years group, as age was positively associated with total PA, and total PA negatively associated with SES. However, objectively measured PA was positively associated with WHR but had negative association with sex. The waist-to-hip ratio (WHR) is a measure of body fat distribution, and a high WHR indicates a greater amount of abdominal fat. Obesity is linked to an increased prevalence of cardiovascular disease risk factors, which can harm cardiovascular structure and function, and increase the prevalence of most cardiovascular diseases, notably heart failure and coronary heart disease [50]. PA, exercise training, and cardiorespiratory fitness, on the other hand, are associated with significant reductions in most cardiovascular diseases, including heart failure and coronary heart disease [50]. Thus, the results of the study suggest that body fat reduction interventions are warranted in this group of learners.

In the 10- to 12-year-old group, the results of the regression analysis also showed that high self-reported PA was associated with higher age and lower SES, whereas high objectively measured PA was associated with lower HC. The result regarding age might be because children’s activity increases with growth in prepuberty. Furthermore, children from rural areas (lower SES) may be more active because they engage in more physical activity, such as walking to school. The negative association between physical activity and HC is consistent with the previously mentioned association between obesity and PA. In a study by Cozett and colleagues on factors influencing physical activity participation among 11–13-year-old school children in the Western Cape, South Africa, they found that parental and peer influence were the most important predictors of physical activity [51].

The negative association of SB with SES in the 10 to 12-year-old group suggests that children from low socioeconomic backgrounds spend a lot of time in sedentary activities such as watching TV and playing games on cell phones. Coombs and colleagues found an association between television viewing and low socioeconomic status among English children [14], as participants in the highest category socioeconomic class spent less time watching TV than participants in the lowest category socioeconomic class [14].

## 5. Limitation

The authors acknowledge certain limitations of this study. It was a cross-sectional study, so the direction of associations and causality could not be ascertained.

The small number of participants that used pedometer made direct comparison of self-reported and objective PA results difficult.

We were unable to recruit enough participants to meet the required sample size for the pedometer, since the participants frequently misplaced the provided pedometers.

The use of questionnaires is also prone to recall bias. Recall bias was minimised by administering the questionnaires to few randomly selected participants and high degree of consistence was observed.

## 6. Conclusions

This study revealed that a high proportion of primary school learners in Lagos State, Nigeria, have high levels of PA. However, findings also showed children from high socioeconomic background and those with bigger hips are more likely to have low PA while children from low socioeconomic background are likely to spend more time in SB. This implies that government and stakeholders in the health promotion sector should strategize on educating parents in the high socioeconomic class on deliberate actions targeted at improving the PA of their wards. Furthermore, parents in the lower socioeconomic class should be encouraged to work on limiting their children’s a screen time activity of not more than 2 h daily, as research has suggested that longer periods of SB are linked to poor health consequences [13]. The link between watching television or engaging in other forms of leisure screen time and poor health outcomes is often stronger [13]. The result on PA’s inverse association with hip circumference also suggest the need to focus on obesity preventative measures. Finally, the government and other stakeholders in the Nigerian public health sector should implement policies and interventions to promote PA and reduce obesity and educate the public on the importance of healthy and active lifestyles. Efforts must be directed toward the full implementation of the National and Lagos State’s education policy, which mandate regular participation of pre-secondary school leaners in physical and health education (PHE) classes [52,53,54]. Health education can be an inspiration for participation in PA, and provision of quality physical education and a supportive school environment can promote healthy lifestyles, help prevent non-communicable chronic diseases (NCDs) and mental disorders and improve academic performance. Further research on the level of PA in different human developmental stages, including adolescents and adults, should be conducted in different communities in Nigeria.

## Figures and Tables

**Table 1 ijerph-19-10745-t001:** Comparison of participants’ socio-demographic, anthropometric and cardiovascular variables across gender.

CPAQ	Boys = 292	Girls = 251		
	Median (IQR) (LQ, UQ)	Median (IQR)	z	*p*-Value
Age (years)	8.00 (7.00, 9.00)	8.00 (7.00, 9.00)	−0.284	0.776
BMI (kg/m^2^)	16.20 (15.2, 17.30)	15.94 (14.90, 17.20)	−1.679	0.093
Height (m)	1.27 (1.21, 1.32)	1.25 (1.20, 1.32)	−0.983	0.326
Weight (kg)	26.00 (24.00, 29.00)	25.00 (22.00, 29.00)	−1.557	0.120
HC (cm)	63.00 (60.00, 67.00)	64.00 (60.00, 67.00)	−0.619	0.536
WC (cm)	56.00 (54.00, 59.00)	57.00 (54.00, 59.00)	−0.457	0.647
WHR	0.89 (0.87, 0.93)	0.89 (0.86, 0.93)	−0.095	0.925
SES	6.00 (4.20, 7.20)	6.00 (4.80, 7.20)	−0.163	0.870
SBP (mmHg)	117.50 (109.00, 134.00)	115.00 (106.00, 125.00)	−1.910	0.056
DBP (mmHg)	80.00 (70.00, 100.00)	80.00 (66.00, 99.00)	−1.469	0.142
HR (beats/min)	96.00 (82.00, 109.00)	97.00 (87.25, 110.00)	−1.462	0.144
**YAP**	**Boys = 101**	**Girls = 89**		
Age (years)	11.00 (10.00, 11.00)	10.00 (10.00, 11.00)	−1.280	0.201
BMI (kg/m^2^)	17.20 (15.6, 18.65)	17.20 (15.70, 18.15)	−0.561	0.575
Height (m)	1.35 (1.30, 1.39)	1.32 (1.29, 1.39)	−0.653	0.514
Weight (kg)	30.00 (27.00, 34.00)	30.00 (28.50, 32.00)	−0.723	0.470
HC (cm)	65.00 (62.00, 69.00)	65.00 (60.00, 69.00)	−0.423	0.672
WC (cm)	57.50 (55.00, 61.00)	58.00 (55.00, 61.00)	−0.225	0.822
WHR	0.90 (0.86, 0.92)	0.88 (0.85, 0.93)	−0.034	0.973
SES	6.00 (4.20, 7.80)	5.40 (4.20, 6.90)	−2.020	**0.043 ***
SBP (mmHg)	119.00 (110.00, 126.00)	118.00 (110.00, 127.00)	−0.174	0.862
DBP (mmHg)	80 (68.00, 95.00)	76.00 (68.00, 91.00)	−1.194	0.233
HR (beats/min)	90 (80.75, 103.00)	94.00 (87.00, 109.50)	−2.189	**0.029 ***

**HT**—Height, **WT**—Weight, **BMI**—Body mass index, **WC**—Waist circumference, **HC**—Hip circumference, **WHR**—Waist Hip Ratio, SES—Socio-economic status, **SBP**—Systolic blood pressure, **DBP**—Diastolic blood pressure, **HR**—Heart rate, **LQ**—**lower quartile**, **UQ**—Upper quartile. *—significant *p*-value.

**Table 2 ijerph-19-10745-t002:** Comparison of participants’ physical activity and sedentary behaviour scores across gender.

	Boys = 292	Girls = 251	F	*p*-Value
	Median (IQR)	Median (IQR)		
Score	CPAQ (For Children Aged 6–9 Years)			
Sports PA Weekday (min)	180.0 (20.0, 180.0)	180.0 (20.0, 180.0)	0.018	0.892
Sports PA Weekend (min)	300.0 (90.0, 300.0)	300.0 (120.0, 300.0)	2.534	0.112
Leisure PA Weekday (min)	900.0 (135.0, 900.0)	900.0 (160.0, 900.0)	0.471	0.493
Leisure PA Weekend (min)	1350.0 (240.0, 1400.0)	1350.0 (360.0, 1400.0)	0.096	0.757
PA at School Weekday (min)	1080.0 (480.0, 1080.0)	1080.0 (480.0, 1080.0)	1.499	0.221
Total PA Weekday (min)	2130.0 (575.0, 2160.0)	2160.0 (575.0, 2160.0)	0.898	0.344
Total PA Weekend (min)	1650.0 (300.0, 1675.0)	1650.0 (470.0, 1700.0)	1.372	0.242
Total PA (min)	3780.0 (750.0, 3860.0)	3810.0 (785.0, 3870.0)	1.224	0.269
Total PA/day (min)	540.0 (107.1, 551.4)	544.00 (112.14, 552.9)	1.224	0.269
Sedentary Weekday (min)	300.0 (300.0, 360.0)	300.0 (300.0, 360.0)	2.835	0.093
Sedentary Weekend (min)	120.0 (120.0, 180.0)	120.0 (120.0, 180.0)	0.010	0.921
Total Sedentary time (min)	420.0 (420.0, 540.0)	420.0 (420.0, 540.0)	0.001	0.971
Average Sedentary time (min)	60.0 (60.0, 77.14)	60.0 (60.0, 77.14)	0.000	1.000
**YAP (For Children aged 10–12 years)**				
	**n = 101**	**n = 89**		
Activity in School	13.00 (8.00, 15.00)	13.00 (10.00, 15.00)	0.000	0.996
Activity Outside School	12.00 (10.00, 13.00)	12.00 (9.00, 13.00)	0.710	0.401
Total PA School	24.00 (19.00, 29.00)	25.00 (20.00, 28.00)	0.242	0.623
Sedentary Habits	16.00 (14.00, 17.00)	16.00 (14.00, 16.00)	0.114	0.736
**Pedometer (For Children aged 6–9 years)**				
	**n = 83**	**n = 75**	z	*p*-value
Average Step Count	6994.43 (6142.85, 8691.86)	6638.43 (5421.43, 7538.86)	−2.394	**0.017 ***
Total weekly step count	48,961.00 (43,000.00, 60,843.00)	48,469.00 (37,950.00, 52,772.00)	−2.394	**0.017 ***
**Pedometer (aged 10–12 years)**				
	**n = 26**	**n = 17**	z	*p*-value
Average Step Count	6910.71 (5811.86, 9184.18)	6483.29 (5734.21, 8752.71)	−1.056	0.291
Total weekly step count	48,375.00 (40,683.00, 64,149.25)	45,383.00 (45,383.00, 61,269.00)	−1.056	0.291

Note: Comparison of male and female (CPAQ) PA variables was done controlling for BMI Comparison of male and female (YAP) PA variables was done controlling for SES. *—significant *p*-value.

**Table 3 ijerph-19-10745-t003:** Comparison of participants’ levels of physical activity across gender.

		CPAQ (6–9 Years)			
Variable	All	Male	Female	χ^2^	*p*-Value
**PA Level**	n (%)	n (%)	n (%)	1.391	0.238
Met guidelines	441 (87.5)	224_a_ (85.8)	217_a_ (89.3)		
Did not meet guidelines	63 (12.5)	37_a_ (14.2)	26_a_ (10.7)		
**SB Level**	n (%)	n (%)	n (%)	-	-
Met guidelines	504 (100.0)	261 (100)	243 (100)		
Did not meet guidelines	0 (0)	0 (0)	0 (0)		
		**Pedometer (6–9 years)**			
**PA Level**	n (%)	n (%)	n (%)	5.562	**0.018** *
Met guidelines	115 (72.8)	67_a_ (80.7)	48_b_ (64.0)		
Did not meet guidelines	43 (27.2)	16_a_ (19.3)	27_b_ (36.0)		
**SB Level**	n (%)	n (%)	n (%)	6.961	**0.008** *
Met guidelines	138 (87.3)	78_a_ (94.0)	60_b_ (80.0)		
Did not meet guidelines	20 (12.7)	5_a_ (6.0)	15_b_ (20.0)		
		**Pedometer (10–12 years)**			
**PA Level**	n (%)	n (%)	n (%)	0.951	0.329
Met guidelines	29 (67.4)	19_a_ (73.1)	10_a_ (58.8)		
Did not meet guidelines	14 (32,6)	7_a_ (26.9)	7_a_ (41.2)		
**SB Level**	n (%)	n (%)	n (%)	0.993	0.319
Met guidelines	40 (93.0)	25a (96.2)	15a (88.2)		
Did not meet guidelines	3 (7.0)	1a (3.8)	2a (11.8)		

Same subscript letter denotes a subset of gender categories whose column proportions do not differ significantly from each other at the 0.05 level. Note: YAP scores could not be categorised for levels of self-reported PA for children aged 10–12 years because of lack of data for categorisation. *—significant *p*-value.

**Table 4 ijerph-19-10745-t004:** Table 4 shows the results of the correlation matrix for the relationship of PA with participants’ sociodemographic, cardiovascular, and anthropometric variables.

	PA (CPAQ)		SB (CPAQ)		PA (YAP)		SB (YAP)		Pedometer (6–9 Years)		Pedometer (10–12 Years)	
Variable	r	*p*-Value	r	*p*-Value	r	*p*-Value	r	*p*-Value	r	*p*-Value	r	*p*-Value
Age	0.323	**<0.001**	0.015	0.729	0.195	**0.007**	−0.063	0.391	−0.144	0.072	0.113	0.471
BMI	−0.019	0.671	−0.057	0.199	0.280	**<0.001**	0.154	**0.034**	−0.025	0.759	−0.279	0.070
HT	0.092	**0.039**	−0.031	0.490	−0.001	0.991	−0.226	**0.002**	−0.017	0.830	−0.048	0.760
WT	0.057	0.203	−0.031	0.494	0.274	**<0.001**	0.023	0.749	−0.034	0.672	−0.376	**0.014**
HC	0.039	0.384	0.047	0.295	0.348	**<0.001**	−0.003	0.966	−0.080	0.320	−0.498	**0.001**
WC	0.014	0.755	0.067	0.135	0.201	**0.005**	0.015	0.841	−0.021	0.796	−0.275	0.074
WHR	−0.049	0.275	0.008	0.857	−0.250	**0.001**	−0.043	0.559	0.157	**0.049**	0.113	0.472
SES	−0.318	**<0.001**	−0.005	0.902	−0.176	**0.015**	−0.256	**<0.001**	0.032	0.689	0.130	0.407
SBP	−0.057	0.213	−0.011	0.820	0.041	0.589	−0.119	0.116	0.040	0.634	0.026	0.872
DBP	−0.069	0.133	−0.054	0.240	−0.098	0.194	−0.078	0.303	−0.075	0.375	0.073	0.645

**Abbreviations HT**—Height, **WT**—Weight, **BMI**—Body mass index, **WC**—Waist circumference, **HC**—Hip circumference, **WHR**—Waist Hip Ratio, SES—Socio-economic status, **SBP**—Systolic blood pressure, **DBP**—Diastolic blood pressure, **HR**—Heart rate. *—significant *p*-value.

**Table 5 ijerph-19-10745-t005:** Predictors of self-reported and objective PA among school children’s participants.

Model 1: Dependent variable: Total PA (CPAQ) (6–9 years)					
	B	Std. Error	Beta	t	*p*-value
(Constant)	787.071	808.770		0.973	0.331
Age (years)	455.386	50.360	0.425	9.043	**<0.001 ***
Height (m)	−440.126	736.778	−0.028	−0.597	0.551
SES	−201.383	34.356	−0.232	−5.862	**<0.001 ***
**Model 2: Dependent variable: Total PA (YAP) (10–12 years)**					
(Constant)	−0.597	31.110		−.019	0.985
Age (years)	1.638	0.621	0.183	2.639	**0.009 ***
BMI (kg/m^2^)	0.377	0.265	0.134	1.424	0.156
Weight (kg)	0.057	0.169	0.038	0.335	0.738
WC	0.105	0.506	0.069	0.208	0.835
HC	0.183	0.443	0.152	0.413	0.680
WHR	−15.728	34.085	−0.105	−0.461	0.645
SES	−1.000	0.251	−0.271	−3.988	**<0.001 ***
**Model 3: Dependent variable: Total Step Count (6–9 years)**					
(Constant)	−1187.180	27,586.189		−0.043	0.966
WHR	66,090.243	30,541.565	0.169	2.164	**0.032 ***
Sex	−5533.410	2478.773	−0.174	−2.232	**0.027 ***
**Model 4: Dependent variable: Total Step Count (10–12 years)**					
(Constant)	126,016.433	23,840.560		5.286	**<0.001 ***
Weight (kg)	160.107	651.489	0.047	0.246	0.807
HC (cm)	−1269.125	506.600	−0.484	−2.505	**0.017 ***
**Model 5: Dependent variable YAP SB Scores (10–12 years)**					
Constant	20.079	3.878		5.178	**<0.001 ***
BMI	0.082	0.070	0.083	1.173	0.242
HT	−3.437	2.712	−0.094	−1.268	0.207
SES	−0.282	0.095	−0.218	−2.969	**0.003 ***

**Abbreviations HT**—Height, **WT**—Weight, **BMI**—Body mass index, **WC**—Waist circumference, **HC**—Hip circumference, **WHR**—Waist Hip Ratio, **SES**—Socio-economic status. *—significant *p*-value.

## Data Availability

The datasets generated from this study have been reported in this paper. Due to privacy and ethical concerns, the datasets could not be made publicly available.

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
