# Peer review of "Levels and Patterns of Physical Activity and Sedentary Behaviour of Primary School Learners in Lagos State, Nigeria"

_ijerph, 2022, doi:10.3390/ijerph191710745_

Round 1

Reviewer 1 Report

Dear Authors,

thank you for the chance to review the manuscript.
Overall, the article is of relevant quality. 
Authors have examined physical activity, sedentary behaviour, and sociodemographic, anthropometric, and cardiovascular correlates of the physical activity
among primary school children in Lagos, Nigeria.
In my opinion, the paper has raised a number of important issues and is worth publishing.
However, it is limited by several shortcomings.
Introduction Please use abbreviations for physical activity (PA) and sedentary behaviour (SB) more consistently.
There is no quotation mark on line 31. Part of the introduction is in italics.
Method Please correct the headings in Table 1 (formatting) and add units (if applicable) in Tables 1 and 2.
Reporting of statistically significant values should be more consistent (bold letters in one place, "*" in another).
Do not put a period in subsection headings.
In a few cases, the numbering of the headings is missing (lines 100, 111, 119,128,137,156).
Discussion and conclusion These sections are generally well written.
I am concerned that you do not use BMI percentiles to assess children. However, in the Conclusion section, you state that "
obesity and living in poor socioeconomic environments are potential risk factors within this subpopulation".
Consider rewriting this statement. First, it is important to explain the risks involved.
Second, the incidence of obesity was not investigated in this study. I feel that this type of statement could mislead the readers.

Author Response

                Point by point responses to Reviewers’ comments

Reviewer’s 1

Manuscript Section

Reviewer’s Comment

Authors’ Response

Abstract

No comments

Introduction

Please use abbreviations for physical activity (PA) and sedentary behaviour (SB) more consistently.
There is no quotation mark on line 31. Part of the introduction is in italics. 

This has been done as suggested by the Reviewer. Revision marked in red.

Methods

No comments

Results

Please correct the headings in Table 1 (formatting) and add units (if applicable) in Tables 1 and 2.
Reporting of statistically significant values should be more consistent (bold letters in one place, "*" in another).
Do not put a period in subsection headings.
In a few cases, the numbering of the headings is missing

This has been done as suggested in tables 1 and 2.

Conclusion

I am concerned that you do not use BMI percentiles to assess children. However, in the Conclusion section, you state that " obesity and living in poor socioeconomic environments are potential risk factors within this subpopulation".
Consider rewriting this statement. First, it is important to explain the risks involved.
Second, the incidence of obesity was not investigated in this study. I feel that this type of statement could mislead the readers.

This statement has been rewritten as suggested by the reviewer. The changes can be found btw line 591 and 600.

Reviewer 2 Report

General overview

This manuscript aims to find patterns of PA and SB in primary school students in Lagos, Nigeria and to identify levels of this items. Secondly, the study intended to examine the sociodemographic, anthropometric, and cardiovascular correlates of PA among this cohort of primary school children. It is an interesting study with several strength points, being a cross-sectional study on 733 students and using standardised procedures.

The manuscript is generally well-written, but the rationale, objectives/hypotheses and methodology are not clearly presented. Also, lack of illustration is pretty obvious and could bore the readers. Given the importance of the subject for general health, the manuscript should give more info on the matter. Although the study results are promising and important for the field of health promotion, I have some major methodological concerns and other issues that the authors need to address before I can accept the manuscript for publication.

Specific comments

Abstract
• Authors should consider adding more info.The Results section does not mention anything about the male/female composition of the group.

• The info should be better presented as to raise the interest of the readers.

• The keywords must be different from the title words.

Introduction

• The information given by the authors is not sufficient to create the background for this important matter to health. Could the authors maybe provide some more background on the importance of informing people early in life about this serious problem they will confront later?

• The rationale for examining this problem should be mentioned more clearly in this section. Why did the authors choose to examine it? At least the reader should be given a background on how important this matter is and informed on the novelty of this study.

• The article innovation should be presented in the Introduction. Describe what the research gap of the paper is and what is new. Please describe the links between the research gap and the goal of the article.

• Lines 81-84 are not enough to make your study of sound scientifical importance. Please provide a subsection here with the title RESEARCH QUESTIONS/HYPOTHESES which would be more convincing about the importance of the study.

Methods

• The number of the questions used in the questionnaire or some examples of them are nowhere to be found.

• It is also necessary to describe and incorporate empirical evidence of validity and reliability of the questionnaire which is applied in this research. This information is essential to assume the soundness of the obtained results.

• It is not clear how the authors applied the study design in terms of a minimum sample size of 384 participants for the study. The pedometers were used on 201 participants. Could this be clarified?

The Results section should be reorganised as to follow each hypothesis/research question or objective. Authors need to write key findings focusing on each one of these after being stated.

An illustration using different colours for items would be helpful in presenting the results as to raise the interest of the readers. The plain explanations preceding the tables are somehow boring and repeated.

In tables 1, 2 and 4 the items are puzzled, seem to be missing or difficult to follow as they are not aligned.

The title proposes patterns that are nowhere to be found in the manuscript.

It would be better to have seen more use of terms like 'originality' and 'significance'. Identify what is new in this study that may benefit readers or how it may advance existing knowledge or create new knowledge on this subject. There should be a clear conclusion on why the research findings are significant for health subject and could be used for the help of those in this situation.

Author Response

                Point by point responses to Reviewers’ comments

Reviewer’s 2

Manuscript Section

Reviewer’s Comment

Authors’ Response

Reviewer’s 2

Abstract

• Authors should consider adding more info. The Results section does not mention anything about the male/female composition of the group.

• The info should be better presented as to raise the interest of the readers.

• The keywords must be different from the title words.

All the comments have been accepted and done as suggested by the Reviewer. Revision shown in the abstract with marker red. More information on male/female has been added.

Introduction

Reviewer’s 2

• The information given by the authors is not sufficient to create the background for this important matter to health. Could the authors maybe provide some more background on the importance of informing people early in life about this serious problem they will confront later?

• The rationale for examining this problem should be mentioned more clearly in this section. Why did the authors choose to examine it? At least the reader should be given a background on how important this matter is and informed on the novelty of this study.

• The article innovation should be presented in the Introduction. Describe what the research gap of the paper is and what is new. Please describe the links between the research gap and the goal of the article.

• Lines 81-84 are not enough to make your study of sound scientifical importance. Please provide a subsection here with the title RESEARCH QUESTIONS/HYPOTHESES which would be more convincing about the importance of the study.

The comments of the reviewer are well noted. Additional background information has been included in this section. This can be found between line 65-69.

The comments of the reviewer are well noted. More information can be found between line 91-95, and line 110- 114.

A subsection has been created for research questions for this study. This can be found between line 119-125.

Reviewer’s 2

Methods

• The number of the questions used in the questionnaire or some examples of them are nowhere to be found.

It is also necessary to describe and incorporate empirical evidence of validity and reliability of the questionnaire which is applied in this research. This information is essential to assume the soundness of the obtained results.

• It is not clear how the authors applied the study design in terms of a minimum sample size of 384 participants for the study. The pedometers were used on 201 participants. Could this be clarified?

The number of items in each of the questionnaire used has been provided in the manuscript. Available information on the psychometric properties of the questionnaires used have also been provided in the manuscript under the questionnaires.

We thank the reviewer for this observation, and it is well noted.  We experienced a situation in which the participants misplaced many of the provided pedometers which almost brought this study to a halt. Only 201 participants wore the pedometer for the entire 7 consecutive days and their data were analyzed. However, we have now highlighted this as a limitation in the manuscript

Results

Reviewer’s 2

The Results section should be reorganized as to follow each hypothesis/research question or objective. Authors need to write key findings focusing on each one of these after being stated.

An illustration using different colours for items would be helpful in presenting the results as to raise the interest of the readers. The plain explanations preceding the tables are somehow boring and repeated.

In tables 1, 2 and 4 the items are puzzled, seem to be missing or difficult to follow as they are not aligned.

The title proposes patterns that are nowhere to be found in the manuscript.

This has been done as suggested by the reviewer.

The items are now well organized and written in different colours to catch the attention of the readers as suggested.

We have restricted our choice of language in this regard to levels of PA and SB to avoid any form of confusion

Reviewer’s 2

Conclusion

There should be a clear conclusion on why the research findings are significant for health subject and could be used for the help of those in this situation.

This has been done as suggested by the reviewer. This can be found between line 591-599.

Round 2

Reviewer 2 Report

Dear Authors,

Thank you for the hard work to review the manuscript.
Overall, the article is of relevant quality. 
Authors have examined physical activity, sedentary behaviour, and sociodemographic, anthropometric, and cardiovascular correlates of the physical activity
among primary school children in Lagos,
Nigeria.

The introduction provides a proper background of the topic. The sections/titles have been improved.

The quality of the images is good enough, but I don’t know if the reviewing version has lower resolution than the final version. If not, images should have better resolution in its final size.

It seems that the English is technically correct.

The experimental design meets the scope of the journal, and it is relevant to the community.
Methods are described detailed enough.

The results and the conclusions are quite interesting and well-discussed.

All data are provided.

The authors have adequately addressed all my comments. I have no further suggestions.

In my opinion, the paper is worth publishing now.